A case against default effect sizes in sport and exercise science

Caldwell Aaron aaron.r.caldwell2.ctr@mail.mil 1 2
Vigotsky Andrew D. 3
1 Thermal and Mountain Medicine, U.S. Army Research Institute of Environmental Medicine , Natick , MA , United States of America
2 Oak Ridge Institute of Science and Education , Oak Ridge , TN , United States of America
3 Departments of Biomedical Engineering and Statistics, Northwestern University , Evanston , IL , United States of America
García-Ramos Amador
Electronic publication date: 2020 Nov 3
Publication date: 2020
Volume: 8
Electronic Location ID: e10314
Received 2020 Jun 1; Accepted 2020 Oct 15
Copyright: ©2020 Caldwell and Vigotsky
Copyright year: 2020
Copyright holder: Caldwell and Vigotsky
License: This is an open access article distributed under the terms of the Creative Commons Attribution License, which permits unrestricted use, distribution, reproduction and adaptation in any medium and for any purpose provided that it is properly attributed. For attribution, the original author(s), title, publication source (PeerJ) and either DOI or URL of the article must be cited.
License URL: https://creativecommons.org/licenses/by/4.0/

Keywords: Appiled statistics, Standardized effect size, Standardized mean difference, Sport science, Exercise science

Funding: The authors received no funding for this particular work.

==============================
Recent discussions in the sport and exercise science community have focused on the appropriate use and reporting of effect sizes. Sport and exercise scientists often analyze repeated-measures data, from which mean differences are reported. To aid the interpretation of these data, standardized mean differences (SMD) are commonly reported as a description of effect size. In this manuscript, we hope to alleviate some confusion. First, we provide a philosophical framework for conceptualizing SMDs; that is, by dichotomizing them into two groups: magnitude-based and signal-to-noise SMDs. Second, we describe the statistical properties of SMDs and their implications. Finally, we provide high-level recommendations for how sport and exercise scientists can thoughtfully report raw effect sizes, SMDs, or other effect sizes for their own studies. This conceptual framework provides sport and exercise scientists with the background necessary to make and justify their choice of an SMD.

Introduction

Effect sizes are a family of descriptive statistics used to communicate the magnitude or strength of a quantitative research finding. Many forms of effect sizes exist, ranging from mean raw values to correlation coefficients. In sport and exercise science, a standardized mean difference (SMD) is commonly reported in studies that observe changes from pre- to post-intervention, and for which units may vary from study-to-study (e.g., muscle thickness vs. cross-sectional area vs. volume). Put simply, an SMD is any mean difference or change score that is divided, hence standardized, by a standard deviation or combination of standard deviations. Thus, even among SMDs, there exist multiple calculative approaches (Lakens, 2013; Baguley, 2009). A scientist must therefore decide which SMD is most appropriate to report for their particular study, or if to report one at all. In this manuscript, we will exclusively be focusing on SMD calculations for studies involving repeated-measures since this is a common feature of sport and exercise science studies; other study designs (i.e., between-subjects) have already been extensively covered elsewhere (Baguley, 2009; Kelley & Preacher, 2012; Hedges, 2008).

Different forms of SMDs communicate unique information and have distinct statistical properties. Yet, some authors in sport and exercise science have staunchly advocated for specific SMD calculations, and in doing so, outright rebuke other approaches (Dankel & Loenneke, 2018). While we appreciate that previous discussions of effect sizes have brought this important topic to the forefront, we wish to expand on their work by providing a deeper philosophical and mathematical discussion of SMD choice. In doing so, we suggest that the choice of an SMD should be based on the objective of each study and therefore is likely to vary from study-to-study. Scientists should have the intellectual freedom to choose whatever statistics are needed to appropriately answer their question. Importantly, this freedom should not be encroached on by broad recommendations that ignore the objectives of an individual scientist. To facilitate these reporting decisions, it is imperative to understand what to report and why.

In this paper, we broadly focus on three things to consider when reporting an SMD. First, before choosing an SMD, a scientist must decide if one is necessary. When making this decision, it is prudent to consider arguments for and against reporting SMDs, in addition to why one should be reported. Second, we broadly categorize repeated-measures SMDs into two categories: signal-to-noise and magnitude-based SMDs. This dichotomy provides scientists with a philosophical framework for choosing an SMD. Third, we describe the statistical properties of SMDs, which we believe scientists should try to understand if they are to report them. We relate these perspectives to previous discussions of SMDs, make general recommendations, and conclude by urging scientists to think carefully about what effect sizes they are reporting and why.

Should I report a standardized mean difference?

Before reporting an SMD—or any statistic for that matter—a researcher should first ask themselves whether it is necessary or informative. When answering this, one may wish to consider arguments both for and against SMDs, in addition to field standards. Here, we briefly detail these arguments, in addition to SMD reporting within sport and exercise science.

Proponents and opponents of standardized effect sizes

Opponents

Although SMDs may be useful in some contexts, they are far from a panacea. Arguments against the use of SMDs, including those by prominent statisticians, are not uncommon. These arguments should be considered when choosing whether or not to report an SMD. In particular, the evidentiary value of reporting an SMD must be considered relative to the strength of the general arguments against SMDs. Below, we have briefly summarized some of the major arguments against the use of SMDs.

As far back as 1969, the use of standardized effect sizes—and by proxy, SMDs—has been heavily criticized. The eminent statistician John Tukey stated that “only bad reasons seem to come to mind” for using correlation coefficients instead of unstandardized regression coefficients to interpret data. To put it simply, scientists should not assume that standardized effect sizes will make comparisons meaningful (Tukey, 1969). This same logic can also be applied to qualitative benchmarks (e.g., Cohen’s d = 0.2 is “small”); we believe it is likely that Cohen would also argue against the broad implementation of these arbitrary benchmarks in all areas of research. Similar arguments against the misuse of standardized effect sizes have been echoed elsewhere (Lenth, 2001; Kelley & Preacher, 2012; Baguley, 2009; Robinson et al., 2003).

Others have outright argued against the use of standardized effect sizes because they oversimplify the analysis of, and distort the conclusions derived from, data. In epidemiology, Greenland et al. (1986) provided a damning indictment of the use of standardized coefficients; namely, because they are largely determined by the variance in the sample, which is heavily influenced by the study design. In psychology, Baguley (2009) offers a similarly bleak view of standardized effect sizes. He argues that the advantages of standardized effect sizes are far outweighed by the difficulties that arise from the standardization process. In particular, scientists tend to ignore the impact of reliability and range restriction on effect size estimates, in turn overestimating the generalizability of standardized effect sizes to wider populations and other study designs (Baguley, 2009).

Proponents

Conversely, prominent statisticians have also argued in favor of standardized effect sizes, especially for facilitating meta-analysis (Hedges, 2008). Cohen (1977) was the first to suggest the use of standardized effect sizes to be useful for power analysis purposes. This is because, unlike the t-statistic, (bias-corrected) standardized effect sizes are not dependent on the sample size. Similarly, while p-values indicate the compatibility of data with some test hypothesis (e.g., the null hypothesis) (Greenland, 2019), SMDs provide information about the ‘effect’ itself (Rhea, 2004; Thomas, Salazar & Landers, 1991). Thus, p-values and t-statistics provide information about the estimate of the mean relative to some test hypothesis and thus are sensitive to sample size, while SMDs strictly pertain to the size of the effect and thus are insensitive to sample size. Moreover, any linear transformation of the data will still yield the exact same standardized effect size (Hedges, 1981). The scale invariance property of a standardized effect size theoretically allows them to be compared across studies, various outcomes, and incorporated into a meta-analysis. Therefore, scientists can measure a phenomenon across many different scales or measurement tools and standardized effect sizes should, in theory, be unaffected. Finally, SMDs can provide a simple way to communicate the overlap of two distributions (https://rpsychologist.com/d3/cohend/).

Comments on Standardized Mean Differences in Sport and Exercise Science

Sport and exercise scientists have also commented on the use of standardized effect sizes (Dankel et al., 2017; Dankel & Loenneke, 2018; Rhea, 2004; Thomas, Salazar & Landers, 1991; Flanagan, 2013). The discussion has focused on the need to report more than just p-values, emphasizing that scientists have to discuss the magnitude of their observed effects. Rhea (2004) also provided new benchmarks for SMDs specific to strength and conditioning research, which is certainly an improvement from just using Cohen’s benchmarks.

If SMDs are to be reported, they should not be done so in lieu of understanding effects on their natural scales. To this end, we agree with the laments of Tukey (1969): too often, standardized effects sizes, particularly SMDs, are relied upon to provide a crutch for interpreting the meaningfulness of results. Default and arbitrary scales, such as “small” or “large” based on those proposed by Cohen (1977), should generally be avoided. SMDs should be interpreted on a scale calibrated to the outcome of interest. For example, Rhea (2004) and Quintana (2016) have demonstrated how to develop scales of magnitude for a specific area of research. When possible, it is best practice to interpret the meaningfulness of effects in their raw units, and in the context of the population and the research question being asked. For example, a 5 mmHg decrease in systolic blood pressure may be hugely important or trivial, depending on the context—here, the SMD alone cannot communicate clinical relevance.

In our opinion, standardized effect sizes can be useful tools for interpreting data when thoughtfully employed by the scientists reporting them. However, sport and exercise scientists should be careful when selecting the appropriate SMD or effect size, and ensure that their choice effectively communicates the effect of interest (Hanel & Mehler, 2019). Herein, we will discuss things to consider when reporting an SMD, and we will close by providing general recommendations and examples that we believe sport and exercise scientists will find useful.

Which standardized mean difference should I report?

To facilitate a fruitful discussion of SMDs, here, we categorize them based on the information they convey. We contend that there are two primary categories of SMDs that sport and exercise scientists will encounter in the literature and use for their own analyses. The first helps to communicate the magnitude of an effect (magnitude-based SMD), and the second is more related to the probability that a randomly selected individual experiences a positive or negative effect (signal-to-noise SMD). These categories serve distinct purposes, and they should be used in accordance with the information a scientist is trying to convey to the reader. We will contrast these SMD categories in terms of the information that they communicate and when scientists may wish to choose one over the other. In doing so, we will show that both approaches to calculating the SMD are distinctly valuable. Finally, we demonstrate that, when paired with background information and other statistics—whether they be descriptive or inferential—each SMD can assist in telling a unique, meaningful story about the reported data.

Signal-to-noise Standardized Mean Difference

The first category of SMDs can be considered a signal-to-noise metric: it communicates the average change score in a sample relative to the variability in change scores. This is called Cohen’s dz, and it is an entirely appropriate way to describe the change scores in paired data. The Z subscript refers to the fact that the comparison being made is on the difference scores (Z = Y − X). This SMD is directly estimating the change standardized to the variation in this response, making it a mathematically natural signal-to-noise statistic.

Cohen’s dz can be calculated with the mean change, δ ¯, and the standard deviation of the changes, σδ, (1) dz=δ ¯σδ.

Alternatively, for convenience, dz can be calculated from the t-statistic and the number of pairs (n), (2) dz=tn.

In Eq. (2), one can see that dz is closely related to the t-statistic. Specifically, the t-statistic is a signal-to-noise metric for the mean (i.e., using its sampling distribution), while dz is a signal-to-noise metric for the entire sample. This means that the t-statistic will tend to increase with increases in sample size, since the estimate of the mean becomes more precise, while (bias-corrected) Cohen’s dz will not change with sample size.

Although Cohen’s dz may be useful to describe the change in a standardized form, it is typically not reported in meta-analyses since it cannot be used to compare differences across between- and within-subjects designs (see SMDs below). It is difficult to interpret the value of this type of SMD; that is, since the signal-to-noise ratio itself is more related to the consistency of a change, one can wonder, how much consistency constitutes a ‘large’ effect? This is in contrast to other types of SMDs, wherein the statistic conveys information about the distance between two central tendencies (mean) relative to the dispersion of the data (standard deviation). Moreover, it appears that, to sport and exercise scientists, the value of this SMD is measuring the degree of the change in comparison to the variability of the change scores (Dankel & Loenneke, 2018). Therefore, scientists’ intent on using dz should consider reporting the common language effect size (CLES) (McGraw & Wong, 1992), also known as the probability of superiority (Grissom, 1994). In contrast to dz, CLES communicates the probability of a positive (CLES > 0.5) or negative (CLES < 0.5) change occurring in a randomly sampled individual (see below).

Alternative to the signal-to-noise Standardized Mean Difference

The information gleaned from the signal-to-noise SMD (Cohen’s dz) can also be captured with the CLES (McGraw & Wong, 1992; Grissom, 1994). In paired samples, the CLES conveys the probability of a randomly selected person’s change score being greater than zero. The CLES is easy to obtain; it is simply the Cohen’s dz (SMD) converted to a probability (CLES = Φ(dz), where Φ is the standard normal cumulative distribution function). Importantly, CLES can be converted back to a Cohen’s dz with the inverse standard normal cumulative distribution function (dz = Φ−1(CLES)). CLES is particularly useful because it directly conveys the direction and variability of change scores without suggesting that the mean difference itself is small or large. Further, current evidence suggests that the CLES is easier for readers to comprehend than a signal-to-noise SMD (Hanel & Mehler, 2019).

Magnitude-based Standardized Mean Difference

The second category of SMDs can be considered a magnitude-based metric: it communicates the size of an observed effect relative to spread of the sample. The simplest and most understood magnitude-based SMD is Glass’s Δ, which is used to compare two groups, and is standardized to the standard deviation of one of the groups. However, a conceptually similar version of Glass’s Δ, which we term Glass’s Δpre, can also be employed for repeated-measures. In Δpre, the mean change is standardized by the pre-intervention standard deviation.1 For basic pre-post study designs, Glasss Δpre is fairly straightforward; mean change is simply standardized to the standard deviation of the pre-test responses. There are other effect sizes for repeated measures designs, such as Cohen’s dav and drm, but for brevity’s sake, these are described in the Appendix. Of note, Δpre, dav, and drm are identical when pre- and post-intervention variances are the same (see Appendix). (3) Δpre=δ ¯σpre

Importantly, Δpre is well-described (Morris & DeShon, 2002; Morris, 2000; Becker, 1988) and can also be generalized to parallel-group designs; in particular, when there are 2 groups, typically a control and treatment group, being compared over repeated-measurements (Morris, 2008). Typically, in these cases, a treatment and control group are being directly compared in a ‘pretest-posttest-control design’ (PPC). A simple version of the PPC-adapted Δpre is (4) Δppc=ΔT−ΔC

where ΔT and ΔC are the Δpre from the treatment and control groups, respectively. There are several other calculative approaches which should be considered for comparing SMDs in a parallel-group designs. We highly encourage further reading on this topic if this type of design is of interest to readers (Morris, 2008; Becker, 1988; Viechtbauer, 2007).

Summary of Standardized Mean Differences

Our distinction between signal-to-noise (namely, Cohen’s dz) and magnitude-based SMDs (including Glass’s Δpre, Cohen’s dav, and Cohen’s drm) provides a conceptual dichotomy to assist researchers in picking an SMD (summarized in Table 1). However, along with the conceptual distinctions, researchers should also consider the the properties of these SMDs. In the following section, we briefly review the math underlying each SMD and its implications. The properties that follow from the math complement the conceptual framework we just presented, in turn providing researchers with a theoretical, mathematical basis for choosing and justifying their choice of an SMD.

Table 1 Types of Standardized Mean Differences for pre-post designs.

Magnitude-based	Glass’s Δpre, Cohen’s dav, Cohen’s drm	
Signal-to-noise	Cohen’s dz	

What are the statistical properties of standardized mean differences?

An SMD is an estimator. Estimators, including SMDs, have basic statistical properties associated with them that can be derived mathematically. From a high level, grasping how an estimator behaves—what makes it increase or decrease and to what extent—is essential for interpretation. In addition, one should have a general understanding of the statistical properties of an estimator they are using; namely, its bias and variance, which together determine the accuracy of the estimator (mean squared error, MSE=Biasθ ˆ,θ2+Varθθ ˆ, for some true parameter, θ, and its estimate, θ ˆ). These properties depend on the arguments used in the estimator. As a result, signal-to-noise and magnitude-based SMDs are not only distinct in terms of their interpretation, but also their statistical properties. Although these properties have been derived elsewhere (e.g., Hedges, 1981; Morris & DeShon, 2002; Morris, 2000; Gibbons, Hedeker & Davis, 1993; Becker, 1988), their implications are worth repeating. In particular, there are several salient distinctions between the properties of each of these metrics, which we will address herein. Although this section is more technical, we will return to a higher-level discussion of SMDs in the next section.

Estimator components

Before discussing bias and variance, we will briefly discuss the components of the formulae and their implications. Of course, all SMDs contain the mean change score, δ ¯, in the numerator, and thus increase linearly with mean change (all else held equal). Since this is common to all SMDs, we will not discuss it further.

More interestingly, the signal-to-noise and magnitude-based SMDs contain very different denominators. To simplify matters, let us assume the pre- and post-intervention standard deviations are equal (σpre = σpost = σ). This assumption is reasonable since pre- and post-intervention standard deviations typically do not substantially differ in sports and exercise science. In this case, the standard deviation of change scores can be found simply: σδ=2σ21−r.

With these assumptions, drm = dav = Δpre for −1 ≤ r < 1, where r is the observed pre-post correlation (Appendix). Greater pre-post correlations, r, are indicative of more homogeneous change scores. This makes the behavior of the magnitude-based SMDs fairly straightforward; that is, the estimates themselves will not be affected by the correlation between pre- and post-intervention scores. Their dependence on σ means that the magnitude-based SMD will blow up as σ → 0. This is in contrast to dz, whose denominator contains both σ and r, making it blow up if either σ → 0 or r → 1 (Fig. 1).

Figure 1 Standardized mean differences for a range of pre-post correlations and pre-intervention standard deviations. Standardized mean differences (SMD) were calculated for a pre-post design study with 20 participants to depict the different properties of the different SMDs. We calculated SMDs for a range of pre-post correlations (r) and pre-intervention standard deviations (σpre), each with a mean change score of 1. Magnitude-based SMDs have similar estimates across the range of pre-post correlations and largely only vary as a function of σpre, whereas signal-to-noise SMDs are a function of both σpre and r. Note, dz blows up as r → 1, and all SMDs blow up as σpre → 0. The standard error of each estimator increases as σpre → 0. Importantly, Δpre has lower or similar standard errors as r → 1, whereas dz has greater standard errors as r → 1. Additional simulations, including those of other SMDs, can be found at 10.17605/OSF.IO/FC5XW.

The parsimonious nature of magnitude-based SMDs arguably makes their interpretation easier; with reasonable assumptions, they only depend on the mean change score and the spread of scores in the sample. On the other hand, when breaking dz down into its constituent parts, it depends on the mean change score, the spread of scores in the sample, and the correlation between pre- and post-intervention scores—the latter two will create σδ. These sensitivities should be understood before implementing an SMD.

Bias

Bias means that, on average, the estimate of a parameter (θ ˆ) differs from the “true” parameter being estimated (θ). Most SMDs follow a non-central t-distribution, allowing the bias to be easily assessed and corrected. As shown by Hedges (1981), SMDs are generally biased upwards with small sample sizes; that is, with smaller samples, SMDs are overestimates of the true underlying SMD (θ ˆ>θ). This bias is a function of both the value of the SMD obtained and the sample size: (5) Ed=d ˆ=dcn−1

(6) ⟹Biasd ˆ,d=d ˆ−d=d1cn−1−1,

where d is the “true” parameter being estimated, d ˆ is its estimate, and cm=1−34m−1 is Hedges’ bias-correction factor (Hedges, 1981) and m = n − 1 is the degrees of freedom for a paired sample. Please note that this degrees of freedom will differ for different study designs and standard deviations. For example, with two groups and a pooled standard deviation, m = n1 + n2 − 2. We have noticed the incorrect use of degrees of freedom in some published papers within sport and exercise science, so we urge authors to be cautious.

Because SMDs are biased, especially in small samples, it is advisable to correct for this bias. Thus, when using Cohen’s d in small sample settings, most sport and exercise scientists should apply a Hedges’ correction to adjust for bias. A bias-corrected d ˆ is typically referred to as Hedges’ g: (7) g=d ˆ⋅cn−1,

where d ˆ can represent any of the SMD estimates outlined above. This correction decreases the SMD by about 10 and 5% with 10 and 15 participants, respectively; corrections are negligible with larger sample sizes. Bias correction can also be applied via bootstrapping (Rousselet & Wilcox, 2019).

More generally, we stress to readers that bias per se is not a bad thing or undesirable property. Especially in multidimensional cases, bias can improve the accuracy of an estimate by decreasing its variance—this is known as Stein’s paradox (Efron & Morris, 1977). Indeed, biased (shrunken) estimators of SMDs have been suggested which may decrease MSE (Hedges & Olkin, 1985). However, these are not commonly employed. Having said this, the upward bias of SMDs is generally a bad thing. As will be discussed in the next subsection, by correcting for the upward bias, we also improve (decrease) the variance of the SMD estimate, in turn decreasing MSE via both bias and variance (Hedges, 1981; Hedges & Olkin, 1985).

Variance

While bias tells us about the extent to which an estimator over- or underestimates the value of a true parameter, variance tells us how variable the estimator is. Estimators that are more precise (less variable) will have tighter standard errors and thus confidence intervals, allowing us to make better judgments as to the “true” magnitude of the SMD.

By looking at formulae for variance and its arguments, we can gain a better understanding of what affects its statistical properties. Below are the variance formulae for Cohen’s dz and Glass’s Δpre, which are the two best understood SMDs for paired designs (Becker, 1988; Gibbons, Hedeker & Davis, 1993; Goulet-Pelletier & Cousineau, 2018; Morris, 2000; Morris & DeShon, 2002). (8) Vardz=n−1nn−31+dz2n−dz2cn−12

(9) VarΔpre=n−1nn−321−r+Δpre2n−Δpre2cn−12

Variances for the biased SMDs (above) can be easily converted to variances for the bias-corrected SMDs by multiplying each formula by c(n − 1)2, which is guaranteed to decrease variance since c(⋅) < 1 (Hedges, 1981).

Each variance formula contains the SMD itself, meaning that variance will tend to increase with an increasing SMD. This also complicates matters for dz; since σδ can increase from a smaller σpre or greater r, dz’s variance explodes with homogeneous populations or change scores (Fig. 1). Such a quality is not very desirable, as typically, we would like more precision as effects become more homogeneous; this property is a further indication that dz is not a measure of effect magnitude. This is in contrast to the magnitude-based SMDs, which become more precise as the effect becomes more homogeneous (Fig. 1). Of note, these differences in variance behaviors do not reflect differences in statistical efficiency; after adjusting for scaling, all are unbiased and equally efficient.

By investigating and understanding the statistical properties of a statistic—here, the SMDs—we can gain a better understanding of what we should and should not expect from an estimate. These properties provide us with an intuitive feel for the implications of the mathematical machinery underlying each SMD, in turn helping us choose and justify an SMD.

Considering Previous Arguments for Signal-to-noise Standardized Mean Differences

There have been arguments against SMDs—at least certain calculative approaches—with one particular article claiming that magnitude-based SMDs are flawed (Dankel & Loenneke, 2018). Specifically, Dankel & Loenneke (2018) profess the superiority of Cohen’s dz over magnitude-based SMDs—specifically, Glass’s Δpre—because of its statistical properties2 and its relationship with the t-statistic. Regarding the former, Dankel & Loenneke (2018) opine that the magnitude-based SMD is “dependent on the individuals recruited rather than the actual effectiveness of the intervention.” We do not find this to be a compelling argument against magnitude-based SMDs for several reasons. First, it is in no way specific to magnitude-based SMDs; all descriptive statistics are always specific to the sample. Second, if the data are randomly sampled (a necessary condition for valid statistical inference), then the sample should, on average, be representative of the target population. If imbalance in some relevant covariate is a concern, then an analysis of covariance, and the effect size estimate from this statistical model, should be utilized (Riley et al., 2013).

It is certainly the case that Cohen’s dz has a natural relationship with the t-statistic. Stemming from this relationship, Dankel & Loenneke (2018) suggest that it is a more appropriate effect size statistic for repeated-measures designs. Although it is true that dz is closely related to the t-statistic, this does not imply that dz is the most appropriate SMD to report. First, the t-statistic and degrees of freedom (which should be reported) together provide the required information to calculate a Cohen’s dz, meaning dz may contain purely redundant information. Second, although Cohen’s dz has a clear relationship with the statistical power of a paired t-test, we want to emphasize that utilizing an observed effect size in power analyses is an inappropriate practice. Performing such power analyses to justify sample sizes of future work implicitly assumes that (1) the observed effect size is the true effect size; (2) follow-up studies will require this observed SMD; and (3) this effect size is what is of interest (rather than one based on theory or practical necessity). In most cases, observed effect sizes do not provide accurate estimates of the population-level SMD, and utilizing the observed SMD from a previous study will likely lead to an underpowered follow-up study (Albers & Lakens, 2018). Moreover, relying on previously reported effect sizes ignores the potential heterogeneity of observed effect sizes between studies (McShane & Böckenholt, 2014). Rather, there exist alternative approaches to justifying sample sizes (Appendix 2).

In general, and in contrast to Dankel & Loenneke (2018), we believe that SMDs can be used for different purposes—whether to communicate the size of an effect, calculate power, or some other purpose—and what is best for one objective is not necessarily what is best for the others. Furthermore, we want to emphasize that these are not arguments against the use of signal-to-noise SMDs, but rather a repudiation of arguments meant to discourage the use of magnitude-based SMD by sport and exercise scientists.

Recommendations for Reporting Effect Sizes

In most cases, sport and exercise scientists are strongly encouraged to present and interpret effect sizes in their raw or unstandardized form. As others previously discussed, journals should require authors to report some form of an effect size, along with interpretations of its magnitude, instead of only reporting p-values (Rhea, 2004; Thomas, Salazar & Landers, 1991). However, an SMD, along with other standardized effect sizes, do not magically provide meaning to meaningless values. They are simply a convenient tool that can provide some additional information and may sometimes be helpful to those performing meta-analyses or who are unfamiliar with the reported measures. Specifically, there are situations where the outcome measure may be difficult for readers to intuitively grasp (e.g., a psychological survey, arbitrary units from Western Blots, moments of force). In such cases, a magnitude-based SMD—in which the SD of pre- and/or post-intervention measures is used in the denominator—can be used to communicate the size of the effect relative to the heterogeneity of the sample. In other words, a magnitude-based SMD represents the expected number of sample SDs (not the change due to the intervention) by which the participants improve.

Let us consider examples presented previously in the sport and exercise science literature. The examples presented in Fig. 1 by Dankel & Loenneke (2018), in which both interventions have a σpre = 6.05 and undergo a change of δ ¯=3.0 (Δpre=3.06.05=0.5). This can be interpreted simply: the expected change is 0.5 SD units relative to the measure in the sample. Put differently, if the person with the median score (50th percentile) were to improve by the expected change, she would move to the 69th percentile.3 Like a mean change, this statistic is not intended to provide information about the variability of change scores. The magnitude-based SMD simply provides a unitless, interpretable value that indicates the magnitude of the expected change relative to the between-subject standard deviation. Of course, it can be complemented with a standard error or confidence interval if one is interested in the range of values with which the data are also compatible.

The above can be contrasted with Cohen’s dz, which uses the SD of change scores. Again, using the examples presented in Figure 1 of Dankel & Loenneke (2018), Cohen’s dz of 11.62 and 0.25 are reported for interventions 1 and 2, respectively. If one tries to interpret these SMDs in a way that magnitude-based SMDs are interpreted, he will undoubtedly come to incorrect conclusions. The first would suggest that a person with the median score who experiences the expected change would move to >99.99th percentile, and the second would imply that she moves to the 60th percentile. Clearly, both of these interpretations are wrong. As opposed to a magnitude-based SMD, Cohen’s dz is a signal-to-noise statistic that is related to the probability of a randomly sampled individual experiencing an effect rather than its magnitude alone. In our opinion, Cohen’s dz does not provide any more information than that which is communicated by the t-statistic and the associated degrees of freedom (which should be reported regardless of the effect size). Instead, if the signal-to-noise is of interest, a CLES may provide the information a sport and exercise scientist is interested in presenting. Going back to our earlier example (dz = 11.62 and 0.25, respectively), the CLES would be approximately >99% and 59.9%, or the probability of a randomly sample individual undergoing an improvement is >99% or 59.9% for intervention 1 and 2, respectively. As Hanel & Mehler (2019) demonstrated, the CLES may be a more intuitive description of the signal-to-noise SMD. While our personal recommendation leans towards the use of magnitude-based SMDs and CLES, it is up to the individual sport and exercise scientist to decide what effect size they feel is most appropriate for the data they are analyzing and point they are trying to communicate (Hönekopp, Becker & Oswald, 2006).

In choosing an SMD, we also sympathize with Lakens (2013), “ ... to report effect sizes that cannot be calculated from other information in the article, and that are widely used so that most readers should understand them. Because Cohen’s dz can be calculated from the t-value and the n, and is not commonly used, my general recommendation is to report Cohen’s dav or Cohen’s drm.” Along these same lines, if scientists want to present an SMD, it should not exist in isolation. It is highly unlikely that a single number will represent all data in a meaningful way. We believe that data are often best appreciated when presented in multiple ways. The test and inferential statistics (e.g., p-values and t-statistics) should be reported alongside an effect size that provides some type of complementary information. This effect size can be standardized (e.g., Δpre) or unstandardized (raw), and should be reported with a confidence interval (CI). Confidence intervals of a magnitude-based SMD will provide readers with information concerning both the magnitude and compatibility limits of an effect size; CIs can be calculated using formulae, or perhaps more easily, using the bootstrap. In situations where the measurements are directly interpretable, unstandardized estimates are generally preferable. The CLES can also be reported when the presence of a change or difference between conditions is of interest.

Percent changes

It is not uncommon for sport and exercise scientists to report their data using percentages (e.g., percent change). While this is fine if it supplements the reporting of their data in raw units, it can be problematic if it is the only way the data are presented or if the statistics are calculated based on the percentages. In the case of SMDs, an SMD calculated using a percent change is not the same as an SMD calculated using raw units. More importantly, the latter—which is often of greater interest to readers or those performing meta-analysis—cannot be back-calculated from the former. It is imperative that authors consider the properties of the values that they report and what readers can glean from them.

Data sharing

To facilitate meta-analysis, we suggest that authors upload their data to a public repository such as the Open Science Framework, FigShare, or Zenodo (Borg et al., 2020). This ensures that future meta-analysis or systematic reviews efforts have flexibility in calculating effect sizes since there are multitude of possible calculative approaches, designs, and bias corrections (see Baguley (2009)). When data sharing is not possible, we highly encourage sport and exercise scientists to upload extremely detailed descriptive statistics as supplementary material (i.e., sample size per group, means, standard deviations, and correlations), or alternatively, a synthetic dataset that mimics the properties of the original (Quintana, 2020).

Examples

In the examples below, we have simulated data and analyzed it in R (see Supplemental Information) to demonstrate how results from a study in sport and exercise science could be interpreted with the appropriate application of SMDs. For those unfamiliar with R, there is an online web application (https://doomlab.shinyapps.io/mote/) and extensive documentation (https://www.aggieerin.com/shiny-server/introduction/) to simplify the process of calculating SMDs Redundant (Buchanan et al., 2019).

Scenario 1: interpretable raw differences

In the first hypothetical example, let us imagine a study trying to estimate the change in maximal oxygen consumption (V ˙O2; L min−1) in long-distance track athletes before and after a season of training. For this study, maximal V ˙O2 was measured during a Bruce protocol with a Parvomedics 2400 TrueOne Metabolic System. The results of this hypothetical outcome could be written up as the following:

V ˙O2 after a season of training with the track team (mean = 4.13 L min-1, SD = 0.25) increased compared to when they joined the team (mean = 3.89 L ⋅min-1, SD = 0.21), t (7) = 3.54, p = 0.009, δ ¯=0.23 L min-1 95% CI [0.07–0.38]. The CLES indicates that the probability of a randomly selected individual’s V ˙O2 increasing after their first season with the team is 89%.

Scenario 2: uninterpretable raw differences

Now, let us imagine a study trying to estimate the effect of cold water immersion on muscle soreness. For this hypothetical study, muscle soreness is measured on a visual analog scale before and after cold water immersion following a muscle damaging exercise. The muscle soreness score would be represented by cm on the scale measured left-to-right. Because sensations tend to be distributed lognormal (Mansfield, 1974)—and are multiplicative rather than additive—it is sensible to work with the logarithm of the reported soreness levels. Since these logged scores are not directly interpretable, it is sensible to use an SMD to help interpret the change scores. The hypothetical study could be written up as follows:

Muscle soreness was lower after cold water immersion (mean = 27, SD = 7) compared to before (mean = 46, SD = 11) cold water immersion, t (9) = −6.90, p < .001, Glass’s Δpre =  − 2.2 95% CI [−3.2, 1.3]. The CLES indicates that the probability of a randomly selected individual experiencing a reduction in muscle soreness after cold water immersion is 99%.

Conclusion

We contend that the reporting of effect sizes should be specific to the research question in conjunction with the narrative that a scientist wants to convey. In this context, pooled pre- and/or post-study SDs are viable choices for the SMD denominator. This approach provides insight into the magnitude of a given finding, and thus can have important implications for drawing practical inferences. Moreover, the values of this approach are distinct and, in our professional opinion, potentially more insightful than signal-to-noise SMDs, which essentially provide information that is redundant with the t-statistic. At the very least, there is no one-size-fits-all solution to reporting an SMD, or any other statistics for that matter. Despite our personal preference towards other effect sizes, a sport and exercise scientist may prefer a signal-to-noise SMD (dz) and could reasonably justify this decision. We urge sport and exercise scientists to avoid reporting the same default effect size and interpreting them based on generalized, arbitrary scales. Rather, we strongly encourage sport and exercise scientists justify which SMD is most appropriate and provide qualitative (i.e., small, medium, or large effect) interpretations that are specific to that outcome and study design. Also, sport and exercise scientists should be careful to report the rationale for using an SMD over simply presenting raw mean differences. Lastly, the creation of statistical rituals wherein a single statistic, by default, is used to interpret the data is likely to result in poor statistical analyses rather than informative ones (Gigerenzer, 2018). As J.M. Hammersley once warned, “There are no routine statistical questions; only questionable statistical routines” (Sundberg, 1994).

Supplemental Information

Supplemental Information 1 Code to Reproduce Figure

Click here for additional data file.

Supplemental Information 2 Supplemental Instructions for How to Perform the Proposed Analyses

Click here for additional data file.

Supplemental Information 3 Simulation of SMD performance with color coding

Simulated standardized mean differences for a range of pre-post correlations and pre-intervention standard deviations. Standardized mean differences (SMD) were simulated for a pre-post design study with 20 participants to depict the different properties of the different SMDs. To complement the scenarios Dankel and Leonneke depict, we simulated 1,000 studies for a range of pre-post correlations (rho) and pre-intervention standard deviations (sigma-pre). From this figure, one can clearly see that magnitude-based SMDs have similar estimates across the range of pre-post correlations and only vary as a function of sigma-pre, whereas signal-to-noise SMDs are a function of both sigma-pre and rho.

Click here for additional data file.

Supplemental Information 4 Simulation of SMD performance with individual data points

Simulated standardized mean differences for a range of pre-post correlations and pre-intervention standard deviations. Standardized mean differences (SMD) were simulated for a pre-post design study with 20 participants to depict the different properties of the different SMDs. To complement the scenarios Dankel and Leonneke depict, we simulated 1,000 studies for a range of pre-post correlations (rho) and pre-intervention standard deviations (sigma-pre). From this figure, one can clearly see that magnitude-based SMDs have similar estimates across the range of pre-post correlations and only vary as a function of sigma-pre, whereas signal-to-noise SMDs are a function of both sigma-pre and rho.

Click here for additional data file.

We would like to thank Brad Schoenfeld for his feedback during the early drafting of this manuscript; our writing of this paper would not have occurred without his encouragement. In addition, we would like to thank Thom Baguley, Israel Halperin, Keith Lohse, Scott Morris, and Kristin Sainani for their thoughtful feedback.

The opinions or assertions contained herein are the private views of the author(s) and are not to be construed as official or reflecting the views of the Army or the Department of Defense. Any citations of commercial organizations and trade names in this report do not constitute an official Department of the Army endorsement of approval of the products or services of these organizations. No authors have any conflicts of interest to disclose. Approved for public release; distribution is unlimited.

Appendix 1: Standardized Mean Difference Calculative Approaches

Throughout the text, we use Glass’s Δpre as our token magnitude-based SMD. However, there exist other approaches to calculating magnitude-based SMDs. Here, we briefly discuss two other common calculations of magnitude-based SMDs. Of note, these two other calculative approaches may contain some “effects” (variance) from the intervention in the denominator, arguably making Glass’s Δpre a more “pure” (in the sense that the denominator is uncontaminated by intervention effects) magnitude-based SMD.

Cohen’s dav: Some have argued that Cohen’s dz is an overestimate of the SMD, and instead advocate for reporting an SMD very similar to the Cohen’s ds typically utilized for between-subjects (independent samples) designs (Dunlap et al., 1996). The only difference between Cohen’s dav and Cohen’s ds is that the average standard deviation between the two-samples (e.g., pre- and post-intervention assessments in a repeated-measures design) is used rather than the pooled standard deviation. (10) dav=δ ¯σpre+σpost2

Cohen’s drm: The standardized difference between repeated-measures (hence “rm”) is arguably the most conservative SMD among those reported. This approach “corrects” for repeated-measures by taking into account the correlation between the two measurements. (11) drm=δ ¯σpre2+σpost2−2⋅r⋅σpre⋅σpost⋅2⋅1−r

(12) =δ ¯σδ⋅2⋅1−r

(13) =dz⋅2⋅1−r

Appendix 2: Justifying Sample Sizes

There are more appropriate approaches to justifying sample sizes than using previously reported effect sizes. First, if authors have a question that, for some reason, necessitates null hypothesis significance testing, authors should first perform the necessary risk analysis to obtain their desired error rates. Next, authors can specify a smallest effect size of interest (SESOI) or minimal clinically important difference (MCID) (Hislop et al., 2020). Of note, the ontological basis for (or the rationale for the true existence of) such dichotomizations—both in the effect (SESOI, MCID) and p-value domains (α-level)—should be justified. Oftentimes, it is not the researcher, but a reader, clinician, or policymaker who must make a decision; for such decisions, proper, contextual decision analytic frameworks should be employed (Amrhein, Greenland & McShane, 2019; Hunink et al., 2014; Vickers & Elkin, 2006). Second, if relying on estimation rather than hypothesis testing, sport and exercise scientists could determine a sample size at which they would have high enough “assurance” that the estimates would be sufficiently accurate (i.e., confidence intervals around the effect size are sufficiently narrow) (Maxwell, Kelley & Rausch, 2008). Third, authors may simply be working under constraints (e.g., time, money, or other resources) that prohibit them from recruiting more than n participants. We believe such pragmatic constraints are perfectly reasonable and justifiable. No matter the sample justification, it should be thoughtful and reported transparently. Importantly, the utility of an effect size or SMD should not be determined by its ability to be used in sample size justifications or calculations.

Additional Information and Declarations

Competing Interests

Author Contributions

Data Availability

1 Although conceptually similar, Glass’s Δ and Δpre have different distributional properties (Becker, 1988).

2 Dankel & Loenneke (2018) suggest that, “ ... normalizing effect size values to the pre-test SD will enable the calculation of a confidence interval before the intervention is even completed ... This again also points to the flaws of normalizing effect sizes to the pretest SD because the magnitude of the effect ... is dependent on the individuals recruited rather than the actual effectiveness of the intervention” (p. 4). This is of course not the case, since the variance of the SMD will depend on, among other things, the change scores themselves. Thus, the confidence interval of the magnitude-based SMD estimate cannot be calculated a priori.

3 Assumes a normal distribution. We note that Dankel & Loenneke (2018) data vignettes are approximately uniformly distributed which is an odd assumption to make about theoretical data, but nonetheless, sufficiently conveys the point.

The authors declare there are no competing interests.

Aaron Caldwell conceived and designed the experiments, performed the experiments, analyzed the data, authored or reviewed drafts of the paper, drafted supplementary material, and approved the final draft.

Andrew D. Vigotsky conceived and designed the experiments, performed the experiments, analyzed the data, prepared figures and/or tables, authored or reviewed drafts of the paper, and approved the final draft.

The following information was supplied regarding data availability:

R code to reproduce all analyses and figures is available in the Supplemental Files and at Open Science Framework: Caldwell, Aaron R, and Andrew D. Vigotsky. 2020. “The Case Against Default Effect Sizes for Sport and Exercise Science.” OSF. August 24. DOI: 10.17605/OSF.IO/FC5XW.

See also https://osf.io/dp57w/?ref=d15748c0a9be7bef08b3cf0a542f227025cd667c.

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
