# Peer review of "A case against default effect sizes in sport and exercise science"

_PeerJ, doi:10.7717/peerj.10314_

## Round 0.1 · original submission · Minor Revisions

Two experts reviewers have provided useful commments that I believe they could be useful to improve a well-written and well-balanced paper.

·

Basic reporting

Line 29 - Please replace “observing” with “observe”.

Proponents section - While the entire sub-section is well presented and already contains valuable information and a nice small overview, I think that it would also be beneficial to include a very brief discussion regarding p values and moving away from binary thinking regarding the differences between the xy and z, for instance. I saw that you mentioned it in the “Effect size conventions in exercise and sport science” sub-section but I believe that it would add to the story here as well.

Interim section - While I agree with your summary here, don’t you think that it would also be beneficial for our field to consider smallest detectable changes/differences for their outcomes before attempting to answer their research question with a study and then quantify the size of the effect based on how much it exceeds (or how much it doesn’t exceed) a previously defined change that should be considered relevant/important for practitioners/clinicians in our field. In this way, qualitative descriptors would not be as important and the readers would be “forced” to interpret the results in a way that relates to their own practice. This would generally be a push toward interpreting the data on an individual basis (i.e., what’s meaningful for me, for my context, as it might (or should) differ from one scenario/situation/context etc. to the next).

Lines 145:156 - Please amend this sentence "The Z subscript refers to the difference being compared is no longer between the measurements (X or Y ) but the difference (Z = Y − X )" so that future readers can relate to it better.

Line 151 - Please add “that” between “appears” and “the value”.

Formula (3) - I believe that it would be beneficial, for the sake of comparison and the greater picture, to put Cohen’ds formula as well (with Spooled in the denominator).

Lines 157:159 - I think that this part is a little bit unclear and the readers might have difficulties interpreting what are you trying to say, so I suggest expanding/re-writing this portion of the paragraph. In addition, can you please expand on this topic further, especially on standardised mean changes when a researcher wants to compare the “effectiveness” of the two different interventions. This is something people frequently overlook in their studies, is something that is now increasingly being done in sport and exercise science intervention studies, and is a common issue for meta-analysts looking to compare the effects of two different interventions on a given outcome. I believe it would add a lot to your story here and since you already mentioned the relevant literature that covers it (i.e., Morris and DeShon 2002; Morris, 2008; Becker, 1988; potentially you could check Viechtbauer, 2007), I believe that I won’t take a lot of your time and space in the manuscript.

Line 169 - While I understand that you are trying to be cautious with your statements here, I would still change “it need not exist in isolation” to “it should not exist in isolation”.

Lines 169:179 - Since you are already covering “the reporting” in this sub-section, I would recommend to add a few lines highlighting the importance of “holistic” and accurate (i.e., exact numbers) reporting which will also facilitate more accurate meta-analyses. Essentially, people in sport science field should be better at reporting overall (i.e., not just effect sizes).

Line 194 - There might be a typo regarding “the denominator” and “the numerator”?

Figure 1 - In my opinion, I think you should try to pick a better palette with RColorBrewer. This current figure doesn’t really communicate what it intends since you really have to pay attention to see the difference. This is especially important since, in the next line or so, you said that one can “clearly see”. I actually asked for an opinion of several colleagues on this figure and they thought the same, so I believe its not just me. This is just for the benefit of the communication because I really like the figure even in its current form.

Bias sub-section - When you mention small sample sizes correction, can you please add the recommendation from the literature how small sample size (e.g. 20) should be before needing to apply Hedges’ g correction. Perhaps, you can just put this information in the parentheses where you feel it’s appropriate.

261-262 - While I entirely agree with you here, it is important to note that although the design of randomised trials aims to ensure that baseline characteristics are balanced, imbalance may arise by chance (especially if a trial’s sample size is small) not only through an inadequate randomisation strategy (Riley, 2013). Perhaps, you can add this statement as its relevant for sport science field.

Line 276 - Please flip the numbers in the SMD formula (i.e., current denominator should be numerator and vice versa).

Line 284:296 - Please consider giving a concrete example of reporting the dz effect size, similar to the one above (lines 274:283). You could add that after outlining the wrong way of interpreting dz effect size (lines 290:291). Perhaps, you can also add a statement here that people should actually start reporting t values and associated degrees of freedom as this is rarely the case in our field and potentially causes this “drama” around dz effect size.

Experimental design

No comment

Validity of the findings

No comment

Additional comments

This article was very well-written, and I commend the authors for maintaining a balanced, even tone throughout the piece despite the obvious disagreement with the other research group. I believe that this article facilitates a deeper understanding of different effect sizes (as well as pros and cons of each) which frequently create confusion among researchers regardless of their field of expertise. Therefore, I only have minor suggestions aiming to improve the manuscript further.

·

Basic reporting

Line 36: Elaborate on the specific suggestions/claims the Dankel study makes and move this paragraph to be melded with the fourth paragraph in the introduction, where specific examples are first identified. This is a great manuscript, but the intro didn’t really draw me in. I feel like you could create a bit more interest by detailing the potential issue(s) with the Dankel recommendation.

Experimental design

Line 88: In the current manuscript, authors also make this suggestion. I am not quite sure where in the 1977 Cohen textbook he suggests what is stated in this manuscript, that “standardized effect sizes are not dependent on the sample size”. Similaraly, Dankel and Loenneke suggest: “Importantly, the effect size is supposed to provide information about the statistical test while removing the influence of the sample size”… a page later “Thus, a larger effect size can be obtained by one of 2 ways: (a) a larger mean change from pre to post; or (b) a smaller magnitude of variability with respect to how people responded to the intervention” and attribute this to the supposed ability of dz to negate the effect of sample size, which is easily proven false by doubling any repeated measures data and calculating dz. The mean change will remain and the SD of the change scores will decrease, causing dz to decrease. Unless I am missing something, the math does not support this statement in either the current manuscript or the Dankel article. This issue needs to be addressed somewhere in the manuscript. The following articles may be a place to start.

Negative correlation between sample size and effect size (Slavin and Smith, 2009; https://www.frontiersin.org/articles/10.3389/fpsyg.2019.00813/full


Line 117-130: I think authors are inappropriately over-interpreting the Dankel paper. The Dankel paper really doesn’t say much other than to stop using pooled SDs in repeated-measures analyses ES and power calculations, and provides numerous methods to calculate dz if change SD is not reported. This leaves point 1 in this section irrelevant and distracting. I am fine with point two (the argument against observed effect sizes in power analyses), but I don’t think it is fair to link the whole list to the Dankel paper. Further, if authors are going to make such an argument against observed effect sizes for power analyses, they should provide a more acceptable alternative.


Line 150: “It is difficult to interpret the value of this type of SMD (e.g., how many
standard deviations of the change score is large?).” This is a very tough argument to make, as it could be made for many effect sizes where the raw units are absent. I suggest removing this statement from the manuscript as authors provide a more applied reason to not use this SMD towards the end of the manuscript.


Line 155: It seems like CLES and dz are different ways of conveying the same information and I, again, don’t think it is appropriate to be tough on the Dankel paper here since CLES is a mathematical transformation of dz. However, I do think the authors should describe the potential usefulness of CLES and how it may be a more intuitive than dz.


Line 175: the 2013 Lakens paper states that CIs can be calculated for dz. Thus, I believe this argument should be removed from the manuscript


Line 194: I believe authors meant to say numerator instead of denominator. This should be changed.


Line 200 and : This is a subtly strong statement that sounds like you are suggesting dz is less reliable than other effect sizes, which would require a citation. However, it seems like the more concise statement is that pre-post correlation affects dz, but not other ES calculations. This sentence should be clarified, preferably without using the word, “unstable”.


Line 234-240: See my concerns for Line 200. Again, these statements either need a simulation or citations to show that dz variance is less “stable”. Lakens seems to see nothing wrong with dz in his 2013 article, and seems to suggest it is an accurate representation of an effect size for repeated measures data.

Validity of the findings

Line 245: At this point, readers are aware that the Dankel/Loenneke article is subsumed by this manuscript. This whole section could be removed, and the manuscript would still convey the same information. I suggest removing the whole Previous Arguments section and incorporating lines 257-262 into a much earlier section, closer to page 3.


Line 292: I think this idea should be expanded to include the mean difference confidence interval.


Somewhere in this article, authors should include a bit about journals needing to revisit their statistical requirements, as many journals require the inclusion of effect sizes, which can be an issue for authors or reviewers who may deem effect sizes to be distracting or unproductive for the data at hand.


Effect sizes calculated using percent change data are incompatible with any raw unit effect size. Thus, it is inappropriate to ONLY report percent change mean and SD, as those data are not usable in a meta analysis with other typical effect size data. Thus, the Cohen proponent paragraph is also false, as not all data are compatible as effect sizes.This should be addressed as it is definitely an issue in our field.


Below is a quote from the 2013 Lakens article. I think authors should provide readers with this argument with or without an argument against it. It would probably fit best towards the end of the manuscript.

Lakens, 2013: “I believe this discussion is currently biased by what could be called designism, a neologism to refer to the implicit belief that between-subjects designs are the default experimental design, and that effect sizes calculated from between-subjects designs are more logical or natural. The defense for designism is as follows. It is desirable to be able to compare effect sizes across designs, regardless of whether the observations originate from a within or between-subjects design. Because it is not possible to control for individual differences in between-subject designs, we therefore should consider the effect size that does not control for individual differences as the natural effect size. As a consequence, effect sizes that control for individual differences are “inflated” compared to the “default” (e.g., Dunlap et al., 1996)” … “When empirical questions can only be examined in withinsubjects designs (such as in the case of post-error slowing), effect sizes that control for intra-subjects variability (η2 p and ω2 p), or that take the correlation between measurements into account (Cohen’s dz) are a reasonable statistic to report”

Additional comments

This manuscript provides sport an exercise scientists and practitioners with thoughts and suggestions regarding the use of effect sizes that will probably be novel to most readers. Authors did a great job of gathering and disseminating the effect size knowledge in a relatively easy to digest text. My suggestions do not require major changes, but aim to balance the manuscript as to minimize the chance of reader-perceived bias.

---

## Round 0.2 · Minor Revisions

Please, take into consideration the last comments from the reviewer.

·

Basic reporting

No comment

Experimental design

No comment.

Validity of the findings

No comment.

Additional comments

The authors should be applauded for the great work with this manuscript which will, undoubtedly, facilitate a deeper understanding of different effect sizes commonly used in sport and exercise science. I appreciate the responses of the authors. The changes that the authors have made, based on the suggestions of their colleagues as well as the reviewers, just improved the overall readability of the manuscript and made the message that they wanted to convey more clear. Congratulations!

·

Basic reporting

No further changes suggested

Experimental design

The proponents section should reflect the following:

1) the fact that small sample sizes do in fact affect ALL effect size metrics, including Hedges g. When I triple the following dataset: [pre:1,2,2,3] [post: 2,2,3,4], ANY effect size is around 0.1 larger. Obviously, this effect approaches zero as sample size increases, but it is misleading to continue to perpetuate the idea that effect sizes are not affected by sample size.

2) percent change effect sizes cannot be linearly transformed to non-percent change effect sizes

Validity of the findings

No further changes suggested

---

## Round 0.3 · accepted · Accept

Congratulations for the nice paper and meeting the high standard publication of PeerJ.